# Air Pollution-Induced Neurotoxicity: The Relationship Between Air Pollution, Epigenetic Changes, and Neurological Disorders

**DOI:** 10.3390/ijms26073402

**Published:** 2025-04-05

**Authors:** Sebastian Kalenik, Agnieszka Zaczek, Aleksandra Rodacka

**Affiliations:** 1Department of Oncobiology and Epigenetics, Faculty of Biology and Environmental Protection, University of Lodz, 141/143 Pomorska Street, 90-236 Lodz, Poland; sebastian.kalenik@edu.uni.lodz.pl (S.K.); agnieszka.zaczek@biol.uni.lodz.pl (A.Z.); 2Doctoral School of Exact and Natural Sciences, University of Lodz, 21/23 Jana Matejki Street, 90-237 Lodz, Poland

**Keywords:** air pollution, particulate matter, neurodegenerative diseases, chronic inflammation, oxidative stress, epigenetics

## Abstract

Air pollution is a major global health threat, responsible for over 8 million deaths in 2021, including 700,000 fatalities among children under the age of five. It is currently the second leading risk factor for mortality worldwide. Key pollutants, such as particulate matter (PM_2.5_, PM_10_), ozone, sulfur dioxide, nitrogen oxides, and carbon monoxide, have significant adverse effects on human health, contributing to respiratory and cardiovascular diseases, as well as neurodevelopmental and neurodegenerative disorders. Among these, particulate matter poses the most significant threat due to its highly complex mixture of organic and inorganic compounds with diverse sizes, compositions, and origins. Additionally, it can penetrate deeply into tissues and cross the blood–brain barrier, causing neurotoxicity which contributes to the development of neurodegenerative diseases. Although the link between air pollution and neurological disorders is well documented, the precise mechanisms and their sequence remain unclear. Beyond causing oxidative stress, inflammation, and excitotoxicity, studies suggest that air pollution induces epigenetic changes. These epigenetic alterations may affect the expression of genes involved in stress responses, neuroprotection, and synaptic plasticity. Understanding the relationship between neurological disorders and epigenetic changes induced by specific air pollutants could aid in the early detection and monitoring of central nervous system diseases.

## 1. Introduction

Last year, the State of Global Air Report 2024 was published, which provides a comprehensive analysis of data for air quality and health impacts for countries worldwide in 2021. The report shows that air pollution accounted for 8.1 million deaths globally in 2021. Poor air quality has become the second leading risk factor for death, ahead of tobacco and poor diet, even for children under five years. The disturbing data concern children in particular. They show that in 2021, more than 700,000 deaths in children under 5 years were linked to air pollution; this represents 15% of all global deaths in children under five [1]. The main components of air pollution monitored in most countries are particulate pollution (also known as particulate matter, including PM_2.5_ and PM_10_), ground-level ozone (O_3_), nitric oxide (NO_2_), carbon monoxide (CO), or sulfur oxide (SO_2_). Among them, PM, the main constituent of air pollution, is the most studied due to its massive presence worldwide and being the most harmful to human health. PM is a cocktail of chemicals, metallic components such as lead, titanium, arsenic, and iron, and other components, such as dust and microorganisms, and their toxicity varies depending on the geographical location and the time of year [2]. In some urban agglomerations in Asia, the concentration of PM_2.5_ is many times higher than that in other parts of the globe (https://www.iqair.com, accessed on 1 September 2024). It is also a rule that PM_2.5_ concentrations in urban areas are higher in winter than in different seasons [3]. These fluctuations in the concentration in urban areas may result from various factors, including increased heating during winter, synoptic conditions, or traffic intensity [4].

Epidemiological data indicate that air pollution is an important causative agent of many non-communicable diseases, including asthma, heart disease, stroke, chronic obstructive pulmonary disease, cancer, neurodevelopmental disorders, and congenital disabilities in children [2,5,6,7,8,9,10,11,12,13,14,15,16,17]. Recent epidemiological and animal studies indicate that long-term exposure to air pollution is associated with an increased risk of neurological disorders such as dementia, Alzheimer’s (AD) and Parkinson’s (PD) diseases, multiple sclerosis (MS), and cognitive impairment [18,19,20,21]. In addition, research confirmed that high exposure to air pollution is associated with more rapid rates of cognitive decline over time [7,22,23,24,25,26].

The biological mechanisms responsible for developing neurodegenerative processes induced by exposure to air pollution are still unclear. It is assumed that the induction of neuroinflammation, increased oxidative stress in the central nervous system (CNS), and activation of excitotoxicity are associated with the development of neurological disorders. These processes can, among other things, disrupt the blood–brain barrier and cause epigenetic modifications. The exact temporal sequence of activation remains unclear and warrants further research.

This review presents literature data obtained in epidemiological studies on animal models and in vitro which have demonstrated the influence of air pollution (both gaseous and particulate) on the development of neurodegenerative processes. Much of the article is devoted to epigenetic changes induced by air pollution and their relationship with neurodegenerative disorders. This is a critical issue because it indicates that air pollution through epigenetic modifications can affect the regulation of gene expression, causing excessive activation (e.g., pro-inflammatory genes) or silencing of genes (e.g., performing protective functions). In addition, epigenetic changes can be long-lasting and even inherited by offspring, which can consequently cause the development of diseases many years after exposure or contribute to an increased risk of developing diseases in offspring. Therefore, the offspring’s health depends not only on genetic predispositions recorded in DNA but also on the living environment of the parents.

## 2. Air Pollutants

Air pollutants are a compound blend of gaseous and particulate components. This blend includes complex mixtures of chemical substances. Some compounds are subject to regulatory monitoring due to their known toxic properties; others are omitted. Air quality is currently monitored on an ongoing basis in most countries. The Air Quality Index (AQI) facilitates the interpretation of measurement results. The AQI is calculated differently depending on the region. The most common standards are American and Chinese [27]. The parameters of the AQI defined by the U.S. Environmental Protection Agency (EPA) represent the concentration of five significant pollutants: particulate pollution (also known as particulate matter, including PM_2.5_ and PM_10_), ground-level ozone (O_3_), nitric oxide (NO_2_), carbon monoxide (CO), and sulfur oxide (SO_2_) (https://www.epa.gov/air-quality, accessed on 24 January 2023).

Among the components of air pollution, particulate matter is considered the factor that has the most significant impact on the development of many diseases. The smallest PM particles can penetrate deep into the lungs, enter the bloodstream, and migrate to organs, causing damage to tissues and cells. Gaseous air pollutants contribute to health risks, but their association with increased mortality and morbidity is less well documented [2,27,28,29].

## 3. Gaseous Air Pollutants

### 3.1. Ozone

Ozone (O_3_) is one of the five primary pollutants for which the EPA establishes an AQI. O_3_ is a highly reactive gas that consists of three oxygen atoms from electrical discharges. It is a solid photochemical oxidant, 52% stronger than chlorine [30]. Depending on which layer of the atmosphere it is in, its role and impact on human health and the natural environment is positive or negative. Ozone formed in the stratosphere is “good” because it protects Earth against harmful biologically active UVB radiation (wavelength range 280–320 nm). The depletion of the ozone layer in the stratosphere contributes to increased exposure to UVB, favoring skin disease induction. Anthropogenic pollutants such as freons or nitrogen oxides (mainly N_2_O emissions), resulting from the increasing amounts of nitrogen fertilizers in agriculture, contribute to the depletion of the ozone layer. In turn, ozone in the troposphere (ground-level ozone) is “bad”. It is formed as a result of a series of photochemical reactions of nitrogen oxides and volatile organic compounds (VOCs) emitted by anthropogenic activities [31].

Repeated exposure to ozone from heavily polluted air causes chronic oxidative stress, negatively affecting behavior and cognitive functions [32]. Increased ROS production stimulates the transcription of pro-inflammatory genes and the release of chemokines and cytokines such as IL-1, IL-6, and TNFα, leading to a chronic neuroinflammatory and oxidative stress status that ultimately results in protein accumulation, loss of neurons, immune hypersensitivity, and consequently, cognitive deficits that can trigger neurodegenerative diseases [33,34]. In vivo studies have proven that chronic ozone exposure strongly inhibits rodents’ exploratory behavior and social interaction, negatively affecting short- and long-term memory [35]. According to the literature, ozone causes an increase in lipid peroxidation levels, morphological changes in the nucleus and the cytoplasm, and cell swelling in neurons. It contributes to activated and later phagocytic microglia and increases astrocytes. Moreover, it inhibits the process of neurogenesis in the dentate gyrus in the medial part of the temporal lobe in adult animals, which leads to chronic loss of brain repair in the hippocampus. As a result, in a previous study, rats’ brain changes were analogous to those seen in Alzheimer’s disease [36]. Other studies found that after 24 h ozone exposure, there was a significant decrease in the density of dendritic spines in rats’ olfactory bulb [37]. Others demonstrated a significantly reduced density of dendritic spines in striatal dendrites and pyramidal neurons in the prefrontal cortex, leading to neuronal death and loss of brain repair capacity. In rats, ozone induces motor disturbances, memory deficits, and biochemical changes in brain regions related to memory processes. Farfan-Garcia et al. showed that this was associated with reduced levels of acetylcholine, acetylcholinesterase, and choline acetyltransferase in the CA3 region of the rat hippocampus [32]. The latest studies, published in 2023 and 2024 by Ahmed et al. [38], showed that O_3_ exposure impaired the ability of microglia to associate with and form a protective barrier around Aβ plaques, leading to augmented dystrophic neurites and increased Aβ plaque load.

Although animal studies may help to elucidate potential neurodegenerative mechanisms activated by chronic ozone exposure, there are difficulties in translating the results of these studies to human experiences with neurodegenerative and cognitive disorders. In a literature review published in 2018, Zhao et al. [39] concluded that current evidence for an association between ambient ozone exposure and mental health is inconclusive. Further high-quality research is needed to evaluate potential associations. Recent meta-analyses have shown that ozone increases the risk of the incidence of Parkinson’s disease [40,41,42]. Cohort studies conducted in Canada and published in 2021 also found positive associations between ozone exposure and mortality due to Parkinson’s, dementia, stroke, and multiple sclerosis (MS) [43].

### 3.2. Sulfur Dioxide

Another major air pollutant is sulfur dioxide (SO_2_), which is formed by refining crude oil and during the combustion of sulfur-containing fuels, e.g., coal and crude oil. Other sources of sulfur dioxide include cement production, paper pulp production, metal smelting and processing facilities, and large ships and locomotives burning high-sulfur fuel. Exposure to sulfur dioxide causes shortness of breath, coughing, and exacerbation of cardiovascular and respiratory diseases. Furthermore, epidemiological studies indicate that exposure to SO_2_ is also associated with neurological disorders, such as migraine, stroke, epilepsy, febrile seizure, and brain cancer [44,45]. Animal experiments (mainly rats) have shown that the adverse effect of sulfur dioxide on the central nervous system is associated, among others, with an increase in the expression of pro-inflammatory factors, including tumor necrosis factor-α (TNF-α) and interleukin-1β (Il-1β) in the hippocampus of rats [46]. Sang and others proved that sulfur dioxide inhalation induced neurotoxicity via mechanisms similar to cerebral ischemia, and COX-2-mediated arachidonic acid metabolism, mainly prostaglandin E2 (PGE2) and the functioning of its EP2/4 receptors, played an important role during the process [46].

Subsequent studies by the same team showed that long-term exposure to SO_2_ air pollution at concentrations exceeding environmental standards in rats impaired spatial learning and memory. Researchers found that long-term SO_2_ inhalation can cause synaptic damage, which may be caused by protein kinase A (PKA) and/or protein kinase C (PKC)-mediated signaling pathways [47]. A reduction in the expression of the Arc gene (Activity-regulated cytoskeleton-associated protein) and glutamate receptor subunits (GluR1, GluR2, NR1, NR2A, and NR2B) with a concentration-dependent property in comparison to controls was also observed. An increase in the release of inflammatory cytokines was also confirmed in these animals [45].

An epidemiological study of 704 Alzheimer’s disease patients, which analyzed disease progression in Alzheimer’s disease patients in cities with varying levels of air pollution, showed that high levels of SO_2_ exposure had the most significant impact on cognitive decline in Alzheimer’s disease [48]. Similarly, Chen et al. also found negative associations between SO_2_ and overall cognitive decline and impairment in specific functional areas (including orientation, recall, and language), indicating that SO_2_ has a negative impact on neurocognitive performance [49]. Research also suggests that long-term exposure to SO_2_ is associated with increased rates of multiple sclerosis in children [50].

### 3.3. Nitrogen Oxides

Nitrogen oxides (NOx) are gases produced, among others, as a result of the reaction of nitrous acid or nitric acid with organic materials during the combustion of nitrocellulose (explosives), such as the decomposition of rocket fuel combustion [51]. Nitric oxide (NO) and nitrogen dioxide (NO_2_) are the most dangerous nitrogen oxides, and the level of NO_2_ is used as the AQI for the larger group of nitrogen oxides. Nitrogen oxides are more harmful than sulfur dioxide or carbon monoxide due to their low solubility in water [52]. Namely, gases with high solubility in water are deposited in sections of the upper respiratory tract, resulting in the rapid appearance of poisoning symptoms, i.e., respiratory tract irritation and mucous membranes. As a result, the exposed person takes steps to remove themselves from the exposure site. In turn, gases with low solubility in water, e.g., nitrogen oxides, do not cause symptoms quickly (mucosal irritation). The occurrence of delayed symptoms means that people may be exposed to nitrogen oxides for longer, and these gases can enter the lower respiratory tract [51].

Direct confirmation of the relationships between air NO_2_ exposure and increased risks of cognitive impairment and neurodegenerative diseases was provided by studies in animal models conducted by Yan et al. [53]. Studies have shown that mice NO_2_ inhalation impaired their spatial learning and memory, increased amyloid β42 (Aβ42) accumulation, and promoted pathological abnormalities and cognitive defects associated with Alzheimer’s disease. Microarray and bioinformatics data have shown that the metabolism of arachidonic acid to prostaglandin E2 (PGE2) via cyclooxygenase-2 (COX-2) played a key role in modulating this impairment [53]. NO_2_ exposure may increase the risk of vascular dementia by inducing excitotoxicity in healthy rats and impairing synaptic plasticity directly in stroke-modeled rats [54].

Strong evidence for the induction of neurodegenerative processes as a result of NOx/NO_2_ exposure is provided by population-based cohort studies. Among these studies, it is worth mentioning the study by Chen et al. [7] conducted with 2.1 million people which showed a positive association between NO_2_ and dementia incidence. Subsequent studies confirm that higher exposure to air pollutants, especially NO_2_ and PM_10_, was associated with lower cortical thickness in brain regions affected by AD. It is worth emphasizing that the same studies showed that the increase in greenery indicators (surrounding greenness or amount of green) was associated with greater thickness in the same areas [55]. One of the most recently published studies has shown that the developing brain during childhood and adolescence (ages 9–10 years) is more sensitive to the neurotoxic effects of air pollution (notably higher NO_2_ exposure), with implications for cognitive function and mental health [56]. Meanwhile, a study conducted in Taiwan to assess possible links between prenatal exposure to air pollution and autism spectrum disorder (ASD) found that exposure to CO and NO_2_ during all three trimesters of pregnancy was associated with an increased risk of ASD [57].

A paper published last year focused on investigating the associations between Alzheimer’s disease mortality and socioeconomic factors, some clinical comorbidities, and sources of environmental pollution in Italy. Using methods based on eXplainable AI (XAI), it was found that air pollution (mainly O_3_ and NO_2_) is the most important predictor of Alzheimer’s disease mortality [58].

In recent years, many publications have shown that long-term exposure to ambient air pollution can significantly increase the risk of developing PD. Such studies were carried out, among others, in Denmark and Canada and showed an increased risk of Parkinson’s disease due to increased NO_2_ concentration [43,59]. Other researchers have focused on the spatial dependence of PD occurrence based on air pollution concentrations. An individual spatial analysis conducted in the canton of Geneva, Switzerland, showed that in 6% of patients, the development of PD was not random but followed a spatial dependence. A significant positive association was detected between PD clusters and atmospheric NO_2_ and PM_10_ concentrations [60].

### 3.4. Carbon Monoxide

Due to its physical properties and harmful effects, carbon monoxide (CO) is known as a silent killer. It is an odorless, tasteless, lightweight gas [61]. CO is produced during the incomplete combustion of carbon-based materials, such as coal. Common sources include burning waste (in furnaces), industrial processes, and exhaust gases, mainly from gasoline engines [62]. The affinity of carbon monoxide for hemoglobin is 200–250 times greater than oxygen’s. As a result of carbon monoxide binding with hemoglobin, carboxyhemoglobin (HbCO) is formed, which is incapable of carrying oxygen, leading to hypoxia in tissues and organs, including the brain [63]. Moreover, CO binds with other iron or copper atoms of other metalloproteins in their active sites (e.g., myoglobin, cytochrome c oxidase, cytochrome P450). This further impairs blood oxygenation and, consequently, exacerbates tissue hypoxia [64,65].

Symptoms of carbon monoxide poisoning begin to appear at concentrations of 12%, and at levels exceeding 40%, consciousness disturbances and even death may occur. It should also be noted that in smokers, the level of carboxyhemoglobin can reach 5%, meaning that even brief and low-level exposure to CO can cause symptoms of poisoning [66]. Due to its physical properties, death from CO poisoning is usually accidental. It is most often reported in home or workplace environments. Both single and multiple victims can be affected by carbon monoxide poisoning. Fatal CO poisonings peak during the year’s colder months when heating devices are in use and ventilation systems are closed [61]. Carbon monoxide poisoning affects approximately 50,000 people annually in the United States and presents with a variety of nonspecific symptoms [63]. Patients often do not realize they have been exposed to the gas. The mortality rate from carbon monoxide poisoning ranges from 1% to 3%. Among survivors, between 15% and 40% experience long-term neurological deficits. This results from CO’s impact on mitochondrial processes within cells, leading to disturbances in energy utilization, inflammatory responses, and the production of free radicals that damage nervous tissue, causing cognitive impairment [67].

The central nervous system and heart muscle are most sensitive to hypoxia [64]. Acute exposure to CO is associated with tissue and heart hypoxia, leading to damage and heart attacks, which can further exacerbate hypoxia in the central nervous system, resulting in nervous system damage [63]. Acute brain injury and delayed encephalopathy after acute CO poisoning (DEACMP) are the most common neurological complications following CO poisoning. Additionally, carbon monoxide poisoning can cause a range of disorders, including neurological, psychiatric, and behavioral conditions such as cortical blindness, parkinsonism, amnesia, personality changes, dementia, anxiety, and depression [67]. These neurological complications are thought to result from demyelination in the brain’s white matter. This may be caused by cellular hypoxia due to CO binding to intracellular proteins, neurotoxicity from excessive release of excitatory amino acids like glutamate, or the accumulation of peroxynitrite (ONOO^−^), leading to severe nitrosative stress and subsequent cell death [68].

## 4. Particulate Matter—The Most Dangerous Air Pollutant

Particulate matter is a complex mixture of fine particles and liquid droplets containing organic compounds, acids, metals, soil, dust particles, or semi-volatile compounds. Due to their small size, they pose a serious threat to human health. They can be of natural origin (e.g., fires, volcanic eruptions, sandstorms, or sea salt aerosols) or anthropogenic (artificial), i.e., caused by human activity (e.g., industry, tobacco smoke, car exhaust emissions) [69]. Particle classification is based on the aerodynamic diameter. Among them, three primary groups are distinguished: PM_10_ (particles with a diameter of ≤10 μm), PM_2.5_ (particles with a diameter of ≤2.5 μm), and PM_0.1_, which is also called the ultrafine fraction of particulate matter (UFPM) and have an aerodynamic diameter ≤ 0.1 μm.

Air pollution contains particles such as dust, dirt, soot, and smoke. Larger particles are dark so can be observed directly with the eye. Other particles are too small for optical detection and can only be detected using electron microscopy [70]. The easiest way to visualize the size of PM particles is to compare them to the diameter of human hair. A hair is thought to average around 50–70 μm in diameter, which is approximately seven times larger than PM_10_ and 30 times larger than PM_2.5_ (Figure 1).

Particles with the smallest aerodynamic diameter (<2.5 μm) are the most harmful to health due to their ability to penetrate deep into tissues; they are difficult to remove [71]. They can cross the blood–brain barrier (BBB) and, therefore, pose a threat to brain tissues. They penetrate the olfactory nerves and reach the cerebral cortex and cerebellum, causing oxidative stress and inflammation [72]. Scientific studies clearly show that the ultrafine fraction of particulate matter in macrophages and epithelial cells can increase oxidative stress and damage mitochondria, which was not observed in the case of larger particles [73].

### Routes of PM Entry

The main route of entry of PMs into the body is the respiratory tract. In this system, their fate depends on their size. For the most part, large PM_10_ particles are captured and removed in the upper respiratory tract with the help of an efficient mechanism of the mucociliary apparatus within the mucosa and the cough and sneeze reflex. In turn, in the lower sections of the respiratory tract, smaller pollutants are removed by phagocytosis or by pulmonary macrophages. Solid particles with a diameter of 2.5 μm and smaller get into the alveoli and then directly into the blood vessels, where they can pass through the blood–brain barrier to the brain (Figure 2) [74]. The smallest particles can enter the body through the nasal cavity via the olfactory epithelium or indirectly through the olfactory nerve, trigeminal nerve, and vagus nerve. They thus can spread to other regions, such as the cerebral cortex or cerebellum. Oberdörster et al. [75] concluded that UFPM reaches the brain via the olfactory nerves.

Some particulate matter is swallowed and absorbed through the digestive tract. Total PM concentrations in the gastrointestinal tract may be similar to those in the lungs [76]. Studies have shown that exposure to PMs changed the composition and diversity of the gut microbiome [76]. Kish et al. [77] showed that long-term exposure to PM_10_ altered short-chain fatty acid (SCFA) concentrations and microbiome composition and increased the expression of pro-inflammatory cytokines in the colon. Recent in vitro and in vivo studies have shown that chronic intranasal instillation of PM_2.5_ induces colonic inflammation, dysbiosis in the gut microbiome, and ultimately, brain damage and behavioral changes in mice [78]. The considerable role of intestinal flora in the proper functioning of the human nervous system has been confirmed in a study by Hou et al. [79].

In addition, the eyes can absorb PMs, but this pathway needs to be better understood and studied. Exposure to air pollution containing PM has been demonstrated to cause adverse ocular effects in humans, with PM_2.5_ specifically proven to be toxic to intraocular tissues and contributing to the development of ocular hypertension and glaucoma [80]. In vitro studies indicate that PM_2.5_ exposure affects neural retina (NR) formation in a dose-dependent manner, leading to a reduction in hERO (human embryonic stem cell-derived retinal organoid) size and thickness by suppressing cell proliferation and promoting apoptosis. While differentiation remains largely unaffected, structural abnormalities are observed, including retinal ganglion cell displacement. Transcriptome analysis revealed that PM_2.5_ exposure disrupts the MAPK and PI3K/AKT pathways and reduces fibroblast growth factors (FGF8, FGF10). These findings suggest that PM_2.5_ may impair early retinal development by inhibiting cell proliferation and inducing apoptosis through these molecular pathways [81]. Other studies have linked PM_2.5_ exposure to retinal degeneration, including glaucoma and structural retinal alterations. Chua et al. [82] demonstrated that higher PM_2.5_ exposure correlates with self-reported glaucoma and unfavorable structural features of the disease. Additionally, research indicates that individuals residing in areas with elevated PM_2.5_ absorbance are more likely to experience retinal morphology changes [83]. All of these absorption pathways may promote the development of the pathological mechanisms responsible for neurological and neurodegenerative diseases.

## 5. The Role of Particulate Matter in the Development of Brain Disorders and Diseases

Long-term exposure to PM and UFPM can cause dysfunctions in the nervous system. Research suggests that a small fraction of particles can directly enter the brain through olfactory nerves, generating oxidative stress strongly associated with the pathogenesis of neurodegenerative diseases such as Alzheimer’s and Parkinson’s [84]. Furthermore, direct infiltration of PM into the central nervous system can cause the accumulation of metals (such as titanium, lead, nickel, arsenic, and iron) and other neurotoxic substances in the brain. An example is magnetite nanoparticles, which have been found in human brain tissue [85,86]. Magnetite nanoparticles are small, strongly magnetic iron oxide particles hosting both Fe (III) and Fe (II) in their crystal structure. They are produced during high-temperature combustion and friction processes and form part of the outdoor air pollution mixture. Airborne magnetite pollution particles < ∼200 nm in size can access the brain directly via the olfactory and/or trigeminal nerves, bypassing the blood–brain barrier. One study has reported a significant presence of high-temperature magnetite nanoparticles in temporal cortex samples from Alzheimer’s patients when compared to controls [87]. Excess iron in the cell is toxic to it and may initiate the formation of reactive oxygen species via the catalytic Fenton reaction [88,89]. Studies also suggest a detrimental effect of the magnetic field from magnetite particles on cellular homeostasis and structural changes in proteins, as well as reduced excitability of neurons exposed to a low level of a magnetic field [90,91]. Research confirms that magnetite nanoparticles have a more prominent role in AD than previously thought; their presence was found, for example, in amyloid β plaque cores [92,93,94].

In vitro and in vivo studies on animal models and epidemiological studies show the harmful effects of PM on health in terms of nervous system disorders [95,96,97,98]. Several studies indicate a direct relationship between PM exposure and neurodevelopmental toxicity [99], cognitive and emotional disorders [7,100,101], neurobehavioral alteration [102,103], and diseases related to protein aggregation such as Alzheimer’s and Parkinson’s disease [104,105,106,107,108] (Figure 3).

The susceptibility to neurological effects of particulate matter exposure varies with age [86]. Children and adolescents are particularly vulnerable due to their higher pollutant intake relative to body weight, increased time spent outdoors, and ongoing development of both the nervous and immune systems [109].

Even during the prenatal and perinatal periods, exposure to PM can contribute to altered gene expression patterns associated with neuroplasticity and synaptic function, potentially leading to developmental impairments [110,111]. Epidemiological studies have established links between PM exposure and an increased risk of neurodevelopmental disorders, including attention-deficit hyperactivity disorder (ADHD), cognitive delays, schizophrenia, and autism spectrum disorder [62,112,113,114,115].

Notably, the impact of air pollution on children’s mental health extends beyond neurodevelopmental conditions. Research suggests that exposure to fine particulate matter (PM_2.5_) is associated with a heightened risk of psychiatric disorders, including bipolar disorder, impulse control disorders, and suicidality [44,116,117,118]. In children aged 6–10 years, chronic exposure to high levels of air pollution has also been correlated with emotional and behavioral disturbances, possibly mediated by disruptions in the hypothalamic–pituitary–adrenal (HPA) axis and persistent neuroinflammation [119].

Many cohort studies have shown that chronic exposure to airborne particulate matter not only influences the development of neurodegenerative diseases such as Alzheimer’s disease, Parkinson’s disease, Huntington’s disease (HD), and multiple sclerosis but also significantly accelerates and exacerbates their course [7,19,120,121,122,123,124]. A comprehensive analysis of epidemiological, animal models, and in vitro studies has allowed us to determine the mechanisms of PM-induced neurotoxicity in neurodegenerative diseases. These mechanisms are closely interconnected and contribute to the CNS’s neurotoxic effects.

Many studies have reported that oxidative stress caused by air pollutants plays a key role in CNS damage and is a key modulator in neurodegenerative diseases [125,126,127]. This is because the brain consumes a large amount of oxygen (~20% of the total oxygen used by the body) and is very rich in polyunsaturated fatty acids, which are highly susceptible to ROS. High oxygen consumption leads to excess ROS production. In addition, the brain has low endogenous antioxidant defenses compared to many other tissues (e.g., liver) [128,129]. Many literature studies show that exposure to PM changes lipid profiles, including strongly enhancing lipid peroxidation in the hippocampus and cerebellum [130,131,132]. Fagundes et al. [130] also showed that PM decreased the catalase activity of the hippocampus, cerebellum, striatum, and olfactory bulb. In turn, Lee et al. [126] demonstrated that inhalation of PM_2.5_ resulted in neuronal loss in the cortex, along with elevated levels of phosphorylated tau and the malondialdehyde, oxidative stress marker in the olfactory bulb and hippocampus [126]. In addition, it has been proven that PM_2.5_ inhibition of Nrf2 activity is a key factor for protection against induced oxidative stress [133]. Available evidence suggests that the level of oxidative stress in response to air pollution largely depends on gender, genetic background, and age [134,135,136,137]. Actions directed at inhibiting oxidative stress may prevent PM-induced neurotoxicity. Studies show that antioxidant compounds, including quercetin, vitamin B, and vitamin C, may have protective effects [138,139,140,141].

PM exposure contributes to the activation of glial cells and macrophages in the brain and the induction of neuroinflammation [142]. This is also facilitated by microglial cells’ rapid and intensive uptake of magnetic nanoparticles compared to other nerve cells [143,144]. In addition, activated monocytes from blood can cross the BBB and activate brain microglia [145]. Although neuroinflammation is a neuroprotective mechanism, chronic inflammation is harmful, inhibits neuronal regeneration, and may ultimately lead to neurodegeneration [146,147,148].

Postmortem analyses of brain tissue samples from regions with significant air pollution have revealed increased expression of inflammatory factors and innate immune receptors in many brain areas [149,150,151,152,153,154]. In vivo studies have shown that activation of glial cells induced by PM occurs in brain regions, such as the cortex, cerebellum, midbrain, and hippocampus [96,155,156]. In response to PM, glial cells produce pro-inflammatory cytokines, such as tumor necrosis factor (TNF)-α, interleukin (IL)-1β, IL-16, and chemokines, including the C-C motif chemokine ligand 1 and 2 (CCL1 and CCL2) and IL-18 [96,127,157,158,159,160,161].

In addition to pro-inflammatory mediators, microglial cells also release neurotoxic nitric oxide, which is considered a major factor in reducing the viability of neurons. Kong et al. observed that pro-inflammatory mediators and nitric oxide released from microglia exacerbate neuronal damage, such as synaptic impairment, phosphoric tau accumulation, and neuronal death [96].

Another process influenced by PM, which plays a significant role in inducing neuronal death leading to brain dysfunction, is excitotoxicity (altering neurotransmitter levels). It has been confirmed that exposure to PM induces an increase in glutamate levels in the frontal cortex, hippocampus, and olfactory bulb [162,163] and activates specific NMDA receptor subunits in the hippocampus [136,164]. The role of excessive excitatory neurotransmitter release under the influence of PM was demonstrated by Liang et al. [165] According to the authors, the loss of cognitive functions caused by short-term PM_2.5_ exposure is influenced to a greater extent by excessive stimulation of synapses than by the neuroinflammatory response. Recent studies showed that PM_2.5_ interacts with dopamine receptors, disrupts dopamine signaling by suppressing Drd1 expression, and contributes to the development of mental disorders, including anxiety and depression [166].

Oxidative stress, inflammation, and microglia activation induced by PM are strongly associated with disrupting the blood–brain barrier. Many human and animal studies have shown that PM, upon penetration into the BBB, disrupts its integrity, resulting in increased permeability [151,167,168]. Exposure of pregnant rats to nano-sized traffic-related air pollution (UFPM) resulted in impaired BBB function in male offspring. Analysis of cellular mechanisms showed a 75% decrease in the tight junction protein zonula occludens-1 (ZO-1) in the hippocampus and a twofold increase in iron depositions, a marker of microhemorrhages [115]. Studies on the in vitro BBB model have shown that PM_2.5_ increases the permeability of the BBB layer probably by decreasing the expression level of the ZO-1, the most critical cytoplasmic proteins of tight junctions [96]. Other studies using a similar research model have shown that chronic diesel exhaust particle exposure decreases the expression and function of the membrane efflux transporter P-glycoprotein (P-gp) in the BBB and impairs BBB integrity (i.e., increased permeability). Under physiological conditions, P-gp removes extracellular Aβ from the brain; thus, decreased expression/activity of P-gp may lead to the accumulation of Aβ in the CNS [169]. Recent studies have identified another emerging mechanism involved in the toxicity of PM, namely epigenetic modifications. This issue is presented in the next chapter.

## 6. The Role of Epigenetics in Air Pollution-Induced Neurodegenerative Diseases

Although the contemporary definition of epigenetics has been prevalent for over a decade, the precise mechanisms by which epigenetic modifications affect gene expression are still being elucidated. Unlike Neo-Darwinism [170], which emphasizes random DNA mutations as the primary driver of evolution and discounts the inheritance of acquired characteristics, epigenetics provides a broader understanding of heredity. It is now widely accepted that heritable changes can arise independently of DNA sequence variations. Environmental factors, including pollution, diet, and temperature, can induce phenotypic alterations transmitted across generations via epigenetic inheritance, even after the original exposure ceases. While environmental factors typically do not alter DNA sequences, they can significantly modify the epigenome, influencing processes such as DNA methylation, histone modification, and non-coding RNA expression. These epigenetic changes affect mitotic stability, contributing to phenotypic variability and disease development [170,171]. Environmental epigenetic transgenerational inheritance has profound implications for disease etiology, the inheritance of phenotypic traits, and evolutionary biology, presenting a non-Mendelian mode of inheritance. Moreover, environmental epigenetics and genetics should be viewed as complex, interconnected molecular systems [172].

Studies suggest that chronic exposure to environmental pollutants, encompassing industrial toxins, during both prenatal and postnatal periods, may be a significant factor in the pathogenesis of neurodegenerative diseases. Exposure to specific environmental agents, such as heavy metals (lead, mercury, arsenic), pesticides, and nanoparticles, has been associated with the development of essential neuropathological features characteristic of neurodegenerative disorders, including amyloid plaque formation and neurofibrillary tangle accumulation. Growing evidence supports the hypothesis that environmental pollutants contribute to a range of neurodegenerative and neurological diseases by inducing alterations in gene expression through epigenetic modifications [173,174,175,176,177,178,179,180]. Environmental contaminants can disrupt the intricate regulatory mechanisms that govern gene activity, resulting in changes in mRNA stability, protein synthesis, and epigenetic modifications [181]. Specifically, alterations in DNA methylation patterns, non-coding RNA expression, and histone protein modifications can serve as biomarkers indicative of exposure to neurotoxic environmental contaminants [182,183,184,185]. Understanding the molecular mechanisms by which environmental contaminants trigger epigenetic alterations is essential for developing targeted preventive and therapeutic strategies for neurodegenerative diseases [186] (Figure 4).

Alzheimer’s disease is a neurodegenerative disease primarily affecting the cortex and hippocampus, regions crucial for learning and memory. A hallmark of AD is the accumulation of abnormal proteins, including tangled Tau proteins within neurons (neurofibrillary tangles) and clumps of amyloid beta protein (Aβ) outside neurons (senile plaques) [187]. DNA methylation, a process regulating gene activity, is believed to contribute to the aberrant gene expression patterns observed in AD progression [188]. Studies have shown that lower DNA methylation levels in genes associated with Aβ generation, such as *APP* and *PS1*, are more prevalent in AD patients [189,190]. Additionally, a correlation has been observed between reduced DNA methylation in the *BACE1* promoter, linked to Aβ deposition, and Tau protein neurofibrillary tangles in AD patients [191].

Research suggests that exposure to PM_2.5_, a type of air pollution particle, may induce DNA methylation changes reminiscent of those found in Alzheimer’s disease, including *BACE1, APP, PS1*, and *APOE.* Some studies have demonstrated that PM_2.5_ can influence Aβ generation by altering *BACE1* methylation, leading to a cascade of pathological events [192]. It is hypothesized that PM_2.5_ also affects AD progression through *APOE4* methylation. However, PM_2.5_-induced inflammation and oxidative stress could lead to epigenetic changes and genomic instability. Exposure to PM_2.5_ results in altered DNA methylation at twenty-four CpG sites, each associated with neuropathological markers of Alzheimer’s disease. Notably, several of these CpG sites are situated within genes involved in neuroinflammation, such as *SORBS2*, *PDE11A*, and *GABBR1* [193,194]. However, neuroinflammation induced by PM_2.5_ is mediated by the upregulation of *CD40LG*, *TNF-α*, and *IL-6* and the downregulation of *RBCK1* gene methylation [193,195] (Figure 5).

Autism spectrum disorder, a common neurodevelopmental condition arising in early childhood, is consistently associated with *SHANK3* gene abnormalities and mutations, as well as synaptic development alterations. Rodent models demonstrate that *Shank3* mutations induce ASD-like behaviors, including impaired social interactions. Postmortem brain tissue from individuals with ASD exhibits epigenetic alterations, such as DNA methylation within the CpG-2–4 regions of the *SHANK3* gene [196,197]. Environmental factors, such as PM_2.5_ exposure, significantly impact Shank3 expression, a critical protein for synaptic function. Early-life exposure to PM_2.5_ in neonatal male Sprague Dawley rats decreased Shank3 expression at both mRNA and protein levels in the hippocampus. Furthermore, PM_2.5_ exposure in neuronal cells disrupts gene-specific DNA methylation and mRNA expression of ASD candidate genes, including *Shank3*, which is downregulated due to increased promoter DNA methylation. Moreover, PM_2.5_ exposure in young rats resulted in the development of autistic-like behavioral phenotypes, as evidenced by autism-related behavioral tests, which showed impaired language communication, abnormal repetitive and stereotyped behaviors, social skill deficits, and synaptic abnormalities [196,197] (Figure 5).

Attention-deficit hyperactivity disorder a common neurobehavioral disorder affecting school-aged children, is influenced by both genetic and environmental factors. Genome-wide association studies (GWASs) have identified numerous genes associated with ADHD susceptibility, many of which are involved in synaptic transmission, monoaminergic function, or the catecholaminergic system. Besides genetic factors, environmental influences and gene–environment interactions also contribute to ADHD.

The findings indicate that exposure to ambient polycyclic aromatic hydrocarbons (PAHs) may contribute to behavioral problems associated with ADHD in children [16]. Although PAHs represent one concerning class of neurotoxicants, they are not unique in their potential to disrupt neurodevelopment. Among other risk factors, polychlorinated biphenyls (PCBs) are well studied for their role in neurodevelopmental impairments and their significant impact on ADHD [198]. Research on spontaneous hypertensive rats and PCB-exposed models of ADHD has shown that PCB exposure can alter gene expression patterns in both models, leading to similar ADHD-like symptoms. Genes such as *Gnal, COMT, Adrbk1, Ntrk2, Hk1, Syt11, Csnk1a1, Arrb2, Stx12, Aqp6, Syt1, Ddc, and Pgk1* exhibit altered expression. Significant changes occur in epigenetic modification genes, including *Crebbp*, *Hdac5*, *MeCp2*, and *Dnmt3a*, potentially influencing gene expression through promoter methylation regulation. Hypomethylation of membrane-bound COMT, a schizophrenia and bipolar disorder risk factor, may explain increased COMT expression in PCB-exposed rats. The activation of epigenetic modification genes in spontaneous hypertensive rats suggests adaptive mechanisms [198,199] (Figure 6).

Another environmental factor, bisphenol A (BPA), a ubiquitous plastic and epoxy resin component, disrupts the endocrine system and negatively impacts human health, affecting reproductive, metabolic, brain, and behavioral functions. BPA is linked to neurodevelopmental disorders like ASD and ADHD, as well as cognitive impairments, behavioral disturbances, and neurodegenerative diseases such as Parkinson’s disease, ALS, and multiple sclerosis [198]. Maternal BPA exposure during fetal development in mice leads to hypo- and hypermethylation of CpG islands in brain development and dopaminergic system function genes [200]. Prenatal exposure to bisphenol A at 2500 μg/kg/day (oral gavage) in pregnant Sprague Dawley rats resulted in spatial navigation impairment in their offspring, attributed to the disruption of *Bdnf* gene methylation within both the hippocampus and hypothalamus [201]. Similarly, BPA exposure in mice resulted in significant epigenetic alterations in the cerebral cortex and hippocampus, characterized by decreased 5-mC DNA, altered Dnmt1, Dnmt3a, and HDAC2 enzyme expression, and increased acetylation of histones H3K9 and H3K14, potentially contributing to memory impairments [202] (Figure 6).

Disruptions in synaptic proteins, critical for synapse formation and dendritic spine development, directly impair neuronal function and contribute to cognitive deficits [181]. Consistent with this, Ku et al. [203] found that combined PM_2.5_ and SO_2_ exposure, even at low doses, induced neuronal apoptosis, reduced PSD-95 and NR2B levels, and increased Tau phosphorylation both in vitro and in vivo, highlighting a significant risk for neurodegenerative diseases. These findings highlight the potential risk of neuronal dysfunction and neurodegenerative diseases due to combined exposure to PM_2.5_ and SO_2_ [203]. Furthermore, PM_2.5_ alone has been shown to exhibit neurotoxicity by inhibiting proliferation, inducing apoptosis, generating reactive oxygen species, and altering DNA hydroxymethylation, ultimately disrupting neuronal gene expression and impairing neurite growth and synaptic transmission [204].

Histone acetylation is a dynamic epigenetic process crucial for neural function, regulated by histone acetyltransferases (HATs) and histone deacetylases (HDACs). Studies have linked histone modifications to neurodegenerative diseases such as Alzheimer’s [205]. Environmental pollutant exposure can significantly disrupt this process, potentially contributing to neurotoxicity and disease development. Qiao et al. [206] showed that H3K9 acetylation correlates with stem cell development and neural differentiation.

Conversely, Gjoneska et al. [207] found that decreased H3K27 acetylation in Alzheimer’s mouse brains affected synaptic plasticity genes [207]. Furthermore, modulating histone acetylation via HAT activators or HDAC inhibitors improves memory and synaptic function [33,208,209,210]. These findings suggest that histone modification alterations could serve as early biomarkers of neurotoxic exposure, enabling timely intervention.

H3K9 acetylation, a well-studied histone modification, plays a vital role in regulating genes essential for brain function, including *Bdnf*, *Fos*, and *Arc* [211,212]. Hypoacetylation of H3K9 is associated with various neurological diseases [211,213,214,215]. Environmental pollutants, such as lead, arsenic, nickel, and bisphenol A, impair brain function by decreasing H3K9 acetylation [216,217,218]. Research indicates that reduced H3K9 acetylation correlates with altered expression of *Nos1, Sez6l, Kcna6*, and *Pml* genes, crucial for brain function [219,220,221,222] (Figure 5 and Figure 6).

Studies showed that environmental pollutants such as pesticides (e.g., dieldrin, paraquat) or heavy metals (e.g., nickel) can disrupt the delicate balance of histone acetylation, resulting in reduced acetylation through multiple pathways [223]. Some studies suggest that nickel directly induces histone hypoacetylation, particularly at histone H4 lysine 18. Conversely, others suggest that nickel’s effects are mediated by reactive oxygen species, which can inhibit histone acetyltransferase MOF [179,224,225]. In contrast, studies have also demonstrated that HDAC inhibitors can mitigate the negative effects of nickel exposure, leading to improvements in learning and memory in mice [179]. Furthermore, research by Song indicates that dieldrin exposure increases acetylation of H3 and H4 histone core. Additionally, prolonged (30 days) dieldrin administration in mice induced hyperacetylation of histones within the striatal and substantia nigra regions [226]. In subsequent studies, it was demonstrated that paraquat treatment of dopaminergic cells resulted in histone H3 acetylation due to decreased HDAC activity [227] (Figure 7).

Air pollution significantly alters histone methylation, impacting brain function and contributing to neurodegenerative diseases. An in vivo study showed that chronic PM_2.5_ exposure leads to lower H3K9me2/me3 and higher γ-H2A.X levels in prefrontal white matter nuclei. Decreased H3K9me2/me3 can induce abnormal gene transcription, genomic instability, and DNA damage, resulting in cognitive impairment [228]. Additionally, γ-H2A.X is a marker of DNA double-strand breaks, and its elevated levels suggest potential DNA damage in individuals exposed to polluted air [229]. Exposure to ambient particulate matter, including ultrafine particles, is also associated with neurodevelopmental and neurodegenerative disorders. Recent studies have demonstrated that UFP exposure alters the expression of several non-coding RNAs, which regulates gene expression and fine-tunes cellular responses to environmental stimuli. This alteration may represent a protective mechanism against uncontrolled inflammatory processes, which are known to be neurotoxic. Specifically, UFP-induced changes in non-coding RNA expression might modulate the expression of neuroprotective metallothioneins (MT1A and MT1F) in neurons. Metallothioneins offer protection against aging and heavy metal-induced oxidative stress-related damage and cell death through their anti-inflammatory and antioxidant properties. However, sustained elevated MT1/MTII expression in the brain has been consistently linked to the onset and progression of neurodegenerative diseases such as Alzheimer’s, Parkinson’s, multiple sclerosis, and motor neuron disease [230,231] (Figure 5).

When analyzing the impact of environmental pollution on epigenetic changes, microRNAs (miRNAs) should be considered. These small non-coding RNAs regulate protein expression, and alterations in their profiles, or their loss due to deficiencies in Dicer or DGCR8, have been associated with cognitive decline and neurodegenerative diseases such as Alzheimer’s disease [232]. Exposure to PM, diesel exhaust particles, and carbon black nanoparticles has been shown to disrupt miRNA regulation [233,234].

BACE1, a key enzyme that generates amyloid β peptide by cleaving APP, is activated in early-stage cognitive deficits and plays a crucial role in AD progression. PM_2.5_ has been shown to alter the brain’s inflammatory environment and accelerate AD progression, primarily through BACE1-catalyzed APP cleavage [235]. Ku et al. [236] demonstrated that BACE1 inhibition mitigated the spatial learning and memory deficits associated with PM_2.5_ aspiration. These changes were linked to NF-κB activation, the downregulation of miR-574-5p, and altered binding of miR-574-5p to the 3′UTR of *BACE1*. Reduced miR-574-5p expression, leading to altered binding to the 3′UTR of *BACE1*, may contribute to increased BACE1 levels and subsequent synaptic and cognitive impairment following PM_2.5_ exposure [236] (Figure 5).

Pesticides, such as rotenone, paraquat, and methyl-phenyl-tetrahydropyridine (MPTP), constitute a major class of neurotoxicants linked to neurodegenerative diseases. Studies have suggested that pesticides can affect miRNA expression [237]. In neuroblastoma SH-SY5Y cells, the downregulation of miR-384-5p protects against rotenone-induced neurotoxicity by inhibiting the endoplasmic reticulum stress-regulating protein GPR-78 [237,238]. Other pesticides, like paraquat alone or in combination with Maneb (fungicide), can upregulate miR-195 expression, leading to the downregulation of ADP-ribosylation factors like protein-2 and inducing apoptosis in neural progenitor cells [239]. Additionally, paraquat downregulates miR-200a, inhibiting neural progenitor cell differentiation and increasing beta-catenin protein levels [240] (Figure 7).

In a mouse model of Parkinson’s disease, the neurotoxin MPTP induced miR-494 expression and downregulated the oxidative sensor protein DJ-1 [241]. By enhancing subventricular zone neurogenesis in adult neural stem cells, MPTP increased miR-7 expression and mediated the downregulation of NLRP3 [242]. However, in the MPTP model of PD, the downregulation of BIM by miR-124 regulated apoptosis and autophagy [243] (Figure 6).

The study of brain epigenetics, including changes in DNA methylation, histone modifications, and non-coding RNA, is a promising target for studying the early-life effects of air pollution. The ENVIRONAGE project (ENVIRONmental influence ON AGE-ing in early life) provided evidence of placental molecular processes associated with prenatal air pollution exposure. Alterations in these epigenetic modifications may lead to altered newborn phenotypes and increased susceptibility to developing diseases later in life, including neurodegenerative disorders [244].

The findings and observations suggest that air pollution plays a significant role in the development of neurodegenerative diseases. Moreover, epigenetic changes observed in individuals exposed to pollution can contribute to white matter dysfunction and the progression of neurodegenerative processes.

## 7. Future Perspectives and Conclusions

This review underscores the significant impact of air pollution on neurotoxicity, particularly through mechanisms involving oxidative stress, neuroinflammation, and epigenetic modifications. Among various pollutants, particulate matter (PM_2.5_ and ultrafine particles) has emerged as a key contributor to neurodegenerative diseases, including Alzheimer’s and Parkinson’s disease, as well as cognitive impairments and neurodevelopmental disorders. Further multi-directional research and intensification of changes at various levels are necessary to reduce the adverse effects resulting from excessive environmental pollution. Future studies should focus on elucidating the precise molecular mechanisms underlying pollutant-induced neurotoxicity, particularly the interplay between oxidative stress, neuroinflammation, and epigenetic regulation. Identifying specific biomarkers of air pollution exposure in neurological disorders could facilitate early diagnosis and targeted interventions.

In conclusion, reducing the neurotoxic effects of air pollution improves public health, reduces healthcare costs, increases work efficiency, and stimulates the development of a modern economy. Investing in a cleaner environment is a step toward a healthier, more efficient, and sustainable society.

## Figures and Tables

**Figure 1 ijms-26-03402-f001:**
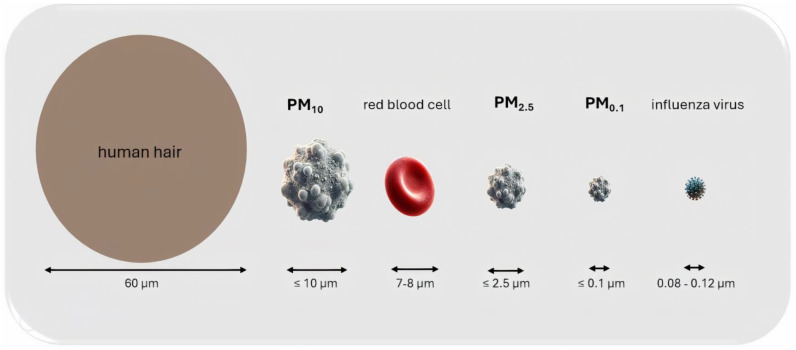
A comparison of particle size to the size of a hair strand, a red blood cell, and the influenza virus.

**Figure 2 ijms-26-03402-f002:**
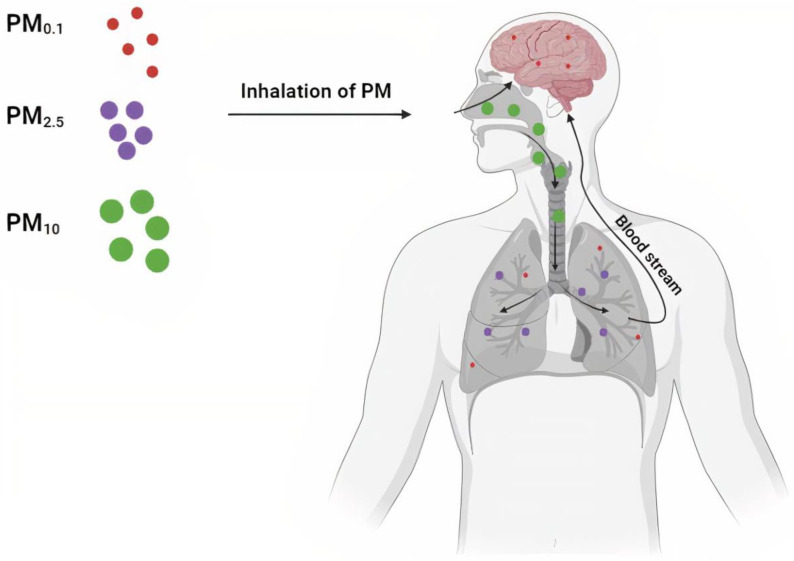
The main areas of deposition of PM. PM_10_ particles (green) are deposited in the upper respiratory tract (nose, throat, larynx), and the fine fractions of both PM_2.5_ (purple) and PM_0.1_ (red) are retained deep in the lungs in the alveoli. The smallest fraction (PM_0.1_)—ultrafine PM—can directly pass through the olfactory epithelium to the CNS. Furthermore, the smallest PM particles can penetrate deep into the lungs and enter the bloodstream, causing tissue and cell damage.

**Figure 3 ijms-26-03402-f003:**
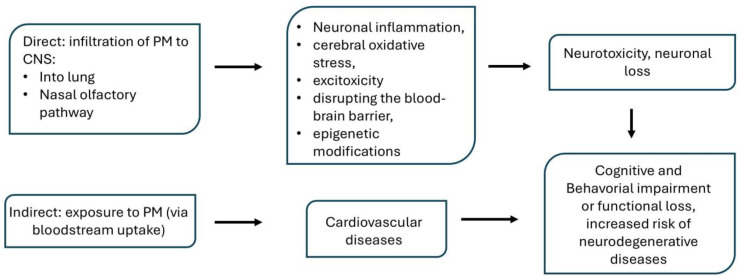
Particulate matter can directly enter the brain, causing neurotoxic effects, and neuronal loss, cognitive deficits, consequently increasing the risk of AD (based on Kilian et al. [73]).

**Figure 4 ijms-26-03402-f004:**
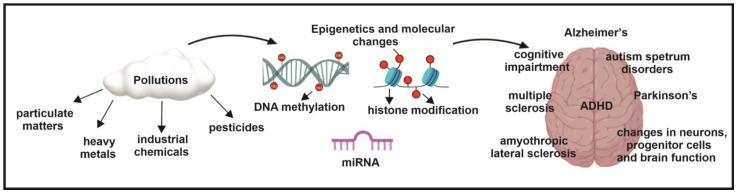
The role of epigenetics in air pollution-induced neurological disorders. Research indicates that air pollutants, including particulate matter, heavy metals, industrial chemicals, and pesticides, can induce epigenetic modifications such as DNA methylation, histone modifications, and miRNA expression. These epigenetic changes are implicated in the pathogenesis of various neurological disorders, including Alzheimer’s disease, cognitive impairment, multiple sclerosis, amyotrophic lateral sclerosis, attention-deficit/hyperactivity disorder, autism spectrum disorders, Parkinson’s disease, disruptions in neuronal function, progenitor cell development, and overall brain function.

**Figure 5 ijms-26-03402-f005:**
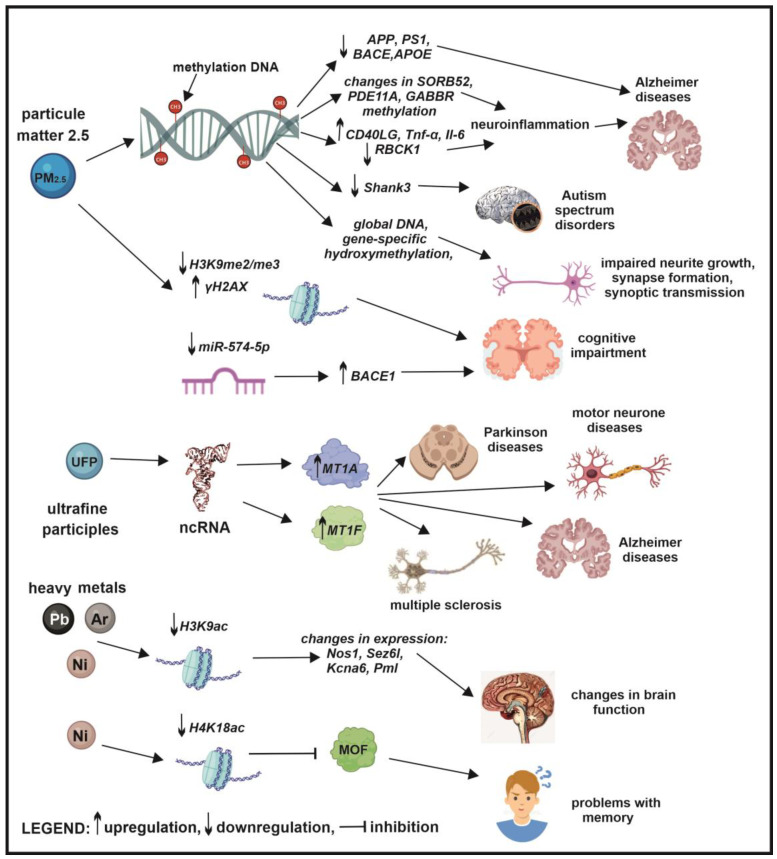
The impact of PM, UFPs, and heavy metals on epigenetic modifications in neurodegenerative diseases. PM exposure induces DNA methylation, histone modifications, and microRNA dysregulation, contributing to the development of Alzheimer’s disease, autism spectrum disorders, impaired growth, synapse formation, synaptic transmission, and cognitive impairment. Similarly, ultrafine particles can regulate non-coding RNAs (ncRNAs), potentially contributing to the pathogenesis of Parkinson’s disease, motor neuron disease, multiple sclerosis, and Alzheimer’s disease. Furthermore, heavy metals, such as lead and mercury, disrupt histone acetylation, leading to alterations in brain development, memory impairments, and other neurological deficits.

**Figure 6 ijms-26-03402-f006:**
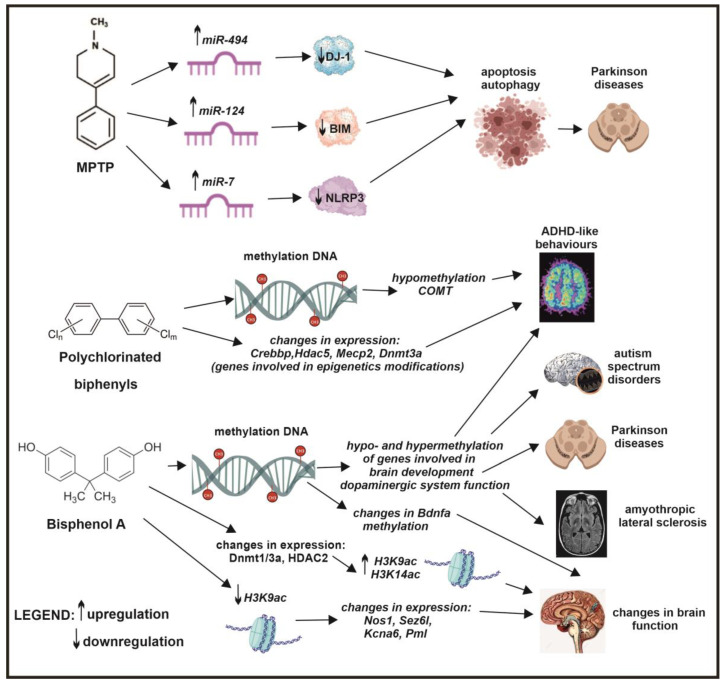
The impact of MTPT, PCBs, and BPA on epigenetic modifications and neurological disorders. MPTP can regulate the expression of specific microRNAs, including miR-494, miR-124, and miR-7, implicated in the pathogenesis of Parkinson’s disease. Polychlorinated biphenyls can alter DNA methylation at genes such as *Crebbp*, *Hdac5*, *Mecp2*, and *Dnmt3a*, potentially contributing to ADHD-like behaviors. Bisphenol A can influence DNA methylation at specific genes and disrupt histone H3 acetylation, potentially associated with the development of autism spectrum disorders, Parkinson’s disease, amyotrophic lateral sclerosis, and alterations in brain function.

**Figure 7 ijms-26-03402-f007:**
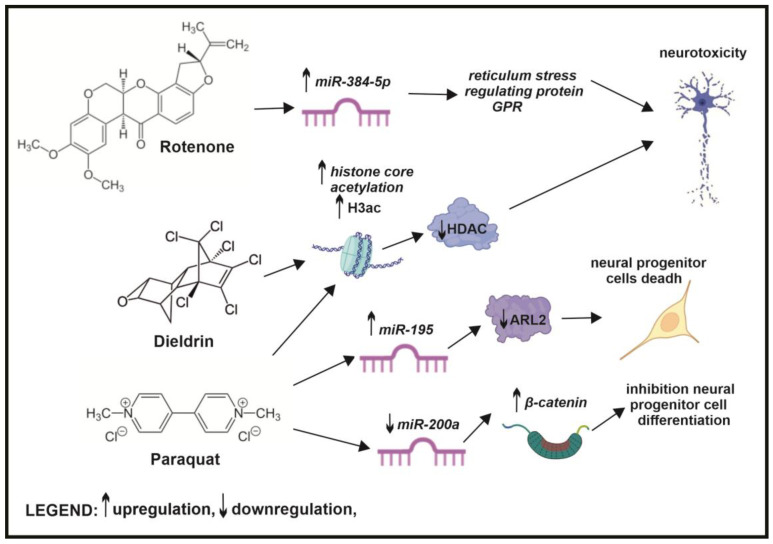
Pesticide-induced epigenetic alterations and the risk of neurological disorders. Pesticides such as rotenone can regulate miR-384-5p expression, while dieldrin can alter histone acetylation, contributing to neurotoxicity. Paraquat can disrupt histone acetylation and change the expression of miR-195 and miR-200a, leading to neural progenitor cell death and inhibiting their differentiation.

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
