# Peer review of "Air Pollution-Induced Neurotoxicity: The Relationship Between Air Pollution, Epigenetic Changes, and Neurological Disorders"

_ijms, 2025, doi:10.3390/ijms26073402_

Round 1

Reviewer 1 Report

Comments and Suggestions for Authors

This manuscript provides a systematic review of the neurotoxic effects induced by air pollutants (including gaseous compounds and particulate matter), with particular emphasis on elucidating the potential epigenetic mechanisms underlying pollution-mediated neurodegeneration. The review demonstrates commendable academic rigor through its well-structured organization. However, prior to publication, the following revisions are recommended to enhance the scholarly value of this work:

  1. The discussion section requires expansion to incorporate a more thorough analysis of pathogenic mechanisms associated with air pollutant exposure.
  2. The authors should provide an in-depth synthesis of emerging evidence regarding epigenetic regulation (including DNA methylation, histone modifications, and non-coding RNA mechanisms) in pollution-induced neurodegenerative pathologies.
  3. The graphical abstracts and schematic diagrams require quality improvement.
  4. Reference numbering must be standardized throughout the text to maintain sequential order (e.g., page 9 line 351; page 9 line 359; page 10 line 380 were not arranged in order).

Author Response

Comments 1: The discussion section requires expansion to incorporate a more thorough analysis of pathogenic mechanisms associated with air pollutant exposure.

Response 1: Thank you for your opinion. We have modified the manuscript to take into account your suggestion. We have expanded some topics to include detailed analysis of mechanisms related to air pollution. Sections 4.1, 5, and 6 have been expanded accordingly.

Comments 2: The authors should provide an in-depth synthesis of emerging evidence regarding epigenetic regulation (including DNA methylation, histone modifications, and non-coding RNA mechanisms) in pollution-induced neurodegenerative pathologies.

Response 2: Thank you for pointing this out. We agree with this comment. Therefore, in the section on epigenetic regulation in neurodegenerative pathologies caused by pollutants, we have thoroughly revised. We have added new information supported by the literature data.

Comments 3: The graphical abstracts and schematic diagrams require quality improvement.

Response 3: We agree with this comment. We have changed all figures improving their quality as suggested.

Comments 4: Reference numbering must be standardized throughout the text to maintain sequential order (e.g., page 9 line 351; page 9 line 359; page 10 line 380 were not arranged in order).

Response 4: Thanks for pointing this out. We agree with this comment. Therefore, we used a bibliographic software package to organize the literature (Mendeley).

Reviewer 2 Report

Comments and Suggestions for Authors

The review "Air Pollution-Induced Neurotoxicity: The Relationship Between Air Pollution, Epigenetic Changes, and Neurological Disorders" by Sebastian Kalenik et al. summarizes the most significant threats of key pollutants inducing neurotoxicity. Overall, the text is readable and the manuscript is of good quality, with a length of 31 pages, 227 references, and 7 figures. The review is interesting, and the number of references corresponds to the length of the text. The overall feel of the manuscript is an overview of key pollutants, such as particulate matter (PM2.5, PM10), ozone, sulfur dioxide, nitrogen oxides, and carbon monoxide, which have substantial adverse effects on human health, contributing to respiratory and cardiovascular diseases, as well as neurodevelopmental and neurodegenerative disorders. The manuscript is suitable for publication, and the recommendations below are for improving the manuscript. The recommendations are for some revisions, which could be minor or medium.

  1. Primarily, the most unpleasant impression from this manuscript is the mess in the listed references in the main body. References have to be numbered in order of appearance in the text and listed individually at the end of the manuscript. It is highly recommended to prepare the references with a bibliography software package, such as EndNote or another. Fixing the citations, according to the publisher's rules is mandatory.
  2. I would recommend to add references for the statements in the following parts of the article:
  • Line 187, Nitrogen oxides are more harmful …
  • Line 255, affects approximately 50,000 people…
  • Line 288, can be detected with an electron microscope. …
  • Line 333-338, In vitro studies have shown…….
  • Line 463, the theory of neo-Darwinism…….
  • Line 506, Tau proteins ….
  • Line 151, Some studies ….
  • Line 546-550, Attention Deficit Hyperactivity Disorder (ADHD)….
  • Line 551, Polychlorinated biphenyls (PCBs) are a well-studied…..
  • Line 598, Studies have linked……
  • Line 627, citation [5] has to be here maybe
  • Line 630, Song's team's studies is not [5] as listed in the references.
  • Line 639, Additionally, γ-H2A.X is a marker of…….
  1. It is highly recommended the parts between lines 376-386 and lines 648-669 be rewritten to avoid the similarity with previous studies.
  2. I would recommend all figures be provided with higher image resolution.
  3. It is extremely difficult to work with the references in this article, and yet there are noticeable omissions of references in the text body such as [8] and [119], and maybe more.

Author Response

Comments 1: Primarily, the most unpleasant impression from this manuscript is the mess in the listed references in the main body. References have to be numbered in order of appearance in the text and listed individually at the end of the manuscript. It is highly recommended to prepare the references with a bibliography software package, such as EndNote or another. Fixing the citations, according to the publisher's rules is mandatory.

Response 1: Thanks you for pointing this out. We agree with this comment. Indeed, we did not follow the journal's citation guidelines. Therefore, we used a bibliography software package (Mendeley) to organize the literature according to the guidelines.

Comments 2: I would recommend to add references for the statements in the following parts of the article:

Response 2: We agree with your comments. We have added literature references to all indicated fragments.

Comments 3: It is highly recommended the parts between lines 376-386 and lines 648-669 be rewritten to avoid the similarity with previous studies.

Response 3: We agree with your comments. We have modified the manuscript as suggested.

Comments 4: I would recommend all figures be provided with higher image resolution.

Response 4: Thanks for pointing this out. We agree with this comment. Therefore we have improved their quality as suggested.

Comments 5: It is extremely difficult to work with the references in this article, and yet there are noticeable omissions of references in the text body such as [8] and [119], and maybe more.

Response 5: Thanks for pointing this out. We agree with this comment. Therefore, we used a bibliographic software package to organize the literature.